# Clinical communication in inflammatory bowel disease: a systematic review of the study of clinician–patient dialogue to inform research and practice

Neda Karimi [1,2], Ria Kanazaki [1,2,3], Annabelle Lukin [4], Alison Rotha Moore [5], Astrid-Jane Williams,[1,2,3] Susan Connor [1,2,3]

► Prepublication history and additional online supplemental material for this paper are available online. To view these files, please visit the journal online (http://dx.doi.org/10.1136/bmjopen-2021-051053).

For numbered affiliations see end of article.

**Correspondence to**
Dr Neda Karimi;
Neda.Karimi@unsw.edu.au

## ABSTRACT

**Objectives**  This systematic review aims to investigate what is currently known about the characteristics of interactions between patients with inflammatory bowel disease (IBD) and their clinicians and its effect on patient outcomes.

**Data sources**  Scopus, PubMed, Embase, Communication Abstracts, Health & Society, Linguistics and Language Behaviour Abstracts and PsycINFO were systematically searched from inception to June 2021.

**Study eligibility criteria**  Peer-reviewed journal articles and book chapters in English investigating the characteristics of naturally occurring interactions between clinicians that manage IBD and patients with IBD during recorded consultations were included.

**Study appraisal and synthesis methods**  Risk of bias was assessed using a specifically developed quality assessment tool, grounded in linguistic theory and the Mixed Methods Appraisal Tool. A narrative synthesis guided by the linguistic concept of metafunction was performed to synthesise the findings.

**Results**  Of the 2883 abstracts reviewed five formed the basis of the review. Interactions between IBD nurses and patients have been mostly characterised in terms of information provision regarding prescribed medications without consideration of the interpersonal aspect. Discussing online medical information with nurses has been shown to improve patient satisfaction. Analyses of gastroenterologist–patient interactions have concentrated on the clinical relationship which has been shown to be disease-centred. Shared decision making in ulcerative colitis has been shown to be compromised due to lack of transparency regarding treatment goals.

**Limitations**  This review did not include articles in languages other than English. Cumulative evidence could not be produced due to the small number of included studies and the diversity of contexts, theories and data types.

**Conclusions and implications of key findings**  There is a paucity of systematic research on naturally occurring clinical communication in IBD and its effect on outcomes. Further research needs to be done to address this knowledge gap.

**PROSPERO registration number**  CRD42020169657.

### Strengths and limitations of this study

► This systematic review summarises and interprets the available evidence on clinical communication in inflammatory bowel disease (IBD) by assessing evidence resulting from investigations of authentic clinical interactions in IBD (real-life data) rather than investigations of self-reported data (perceptions and attitudes).

► The review consults a diverse range of databases, including databases with special focus on medicine, health, psychology, communication and linguistics, to identify eligible studies.

► The synthesis of the results is guided by a well-established theory of language.

► The review does not include articles in languages other than English.

► The review does not provide a complete overview of clinical communication in IBD and its effects due to the availability of limited evidence in this space.

## INTRODUCTION

The main space in which clinicians and patients manage care, negotiate roles and make decisions is their interaction during consultations. Understanding communication between clinicians and patients in this space and its existing variations is crucial for understanding the bigger picture of how disease is managed.

In inflammatory bowel disease (IBD), clinical communication is argued to affect patient satisfaction, treatment adherence, patient quality of life, disease management and self-management.[1–4] In addition, in tandem with the recognition of the importance of minimising disease activity at an early stage in IBD—using a 'treat-to-target' (T2T) management approach,[5 6] research promoting shared decision-making in IBD has gained momentum.[7] A T2T approach involves personalised care and early interventions aimed at delaying or preventing disease

progression, preventing bowel damage and promoting mucosal healing.[6] It is a collaborative approach involving the physician (and the multidisciplinary team) and the patient.[6] It involves joint risk–benefit assessment and decision making, monitoring and optimising therapy to achieve disease control and symptom improvement.[5 8] As a result, discourses around communication and the role of the patient as a key stakeholder in decision making have gained recognition in IBD research.[8]

Currently, studies that investigate communication between IBD clinicians and patients through investigation of the 'clinician–patient interface'[9]—that is, projects that investigate interactions between patients and clinicians, rather than patients' perceptions of clinical communication—are less common. The current review ascertains the existing knowledge in this area in order to inform the field, identify areas that require further investigation, and position this literature within current IBD care practice and research. The main objective is to identify, organise and summarise systematically what is currently known about (1) the characteristics of conversations between clinicians that manage IBD and patients with IBD, and (2) how clinical discussion affects health outcomes in IBD. A secondary objective is to identify the different approaches to the analysis of clinical interaction in IBD, the methodological trends, and the gaps for future research. Underpinning both of these objectives, it is also our aim to show the importance of studying the details of how clinical communication takes place in the specific context of IBD and why drawing only on highly generic principles of healthcare communication or research involving patients with other health conditions is not sufficient for maximising the delivery of quality care in IBD.

## METHOD
### Study selection, data extraction, quality assessment and synthesis
Published peer-reviewed journal articles and book chapters in English that investigated the characteristics of naturally occurring interactions between clinicians that manage IBD and patients with IBD during recorded consultations were included in this review. Self-report studies that explored clinicians' and patients' attitudes, or beliefs about clinical interaction only were excluded. Target participant groups included patients with IBD and clinicians that manage patients with IBD in primary and secondary healthcare (eg, general practitioners, IBD specialists, IBD nurses), complementary and complementary medicine (eg, acupuncturists, traditional Chinese medicine practitioners) or allied health (eg, dietitians). Healthcare providers whose primary treatment includes the interaction itself (eg, psychotherapists) were not included. Studies in which eligible participant groups were present but IBD was not the focus of the study were also excluded.

Scopus, PubMed, Embase, Communication Abstracts, Health & Society, Linguistics and Language Behaviour Abstracts and PsycINFO were searched from inception to June 2021. The reference lists of eligible as well as excluded but relevant publications were screened. Subject matter experts were consulted to ensure inclusiveness. The search strategy is available in our published protocol.[10]

Titles and abstracts were screened by three reviewers (NK, AL and RK). The reviewers identified eligible publications based on their title and abstract, compared their lists of selected publications and resolved any discrepancies prior to the full-text review. Full-text publications were screened for final inclusion by the reviewers with complete agreement between the reviewers.

Data were extracted from included articles by one reviewer (NK) and checked by the review team for accuracy. The template used for data extraction was developed in consultation with the existing health communication and linguistics literature including previous systematic literature reviews of this kind[9 11–14] and included the data items presented in online supplemental table 1.

Risk of bias was assessed using a template designed by the review team in consultation with previous systematic reviews of this kind[13] and compatible with the principles of systemic functional linguistics,[15 16] which offers a conceptual framework for the architecture of language, tools to measure the different components of language, and methodological approaches for principled selection of data and data analysis tools to reduce the description bias and increase credibility. For more information on the assessment of risk of bias refer to the protocol paper.[10] In addition to the tailored template, to assess risk of bias in mixed-methods studies in which the analysis of talk was a secondary aim, the Mixed Methods Appraisal Tool (MMAT)[17] was used.

The extracted data were summarised and compared in overview tables and figures and are interpreted in a narrative synthesis. Halliday's concept of metafunction was used to classify the communicative aspects analysed by the included studies. The concept of metafunction is derived from the view that language has evolved in and through the social contexts in which it functions, and this contextual pressure in its evolution produced three main organising principles for language. All languages, despite their considerable variation, give their speakers resources for 1) construing human experiences (ideational metafunction), 2) enacting personal and social relationships (interpersonal metafunction) and 3) organising a discursive flow and creating cohesion and continuity (textual metafunction).[18] Each linguistic feature analysed by the included papers was coded depending on which of these functions it served. Classifying the included publications based on the metafunction addressed has the advantage of identifying systematically the aspects of IBD clinical communication which have been described as well as those that are not yet explored.

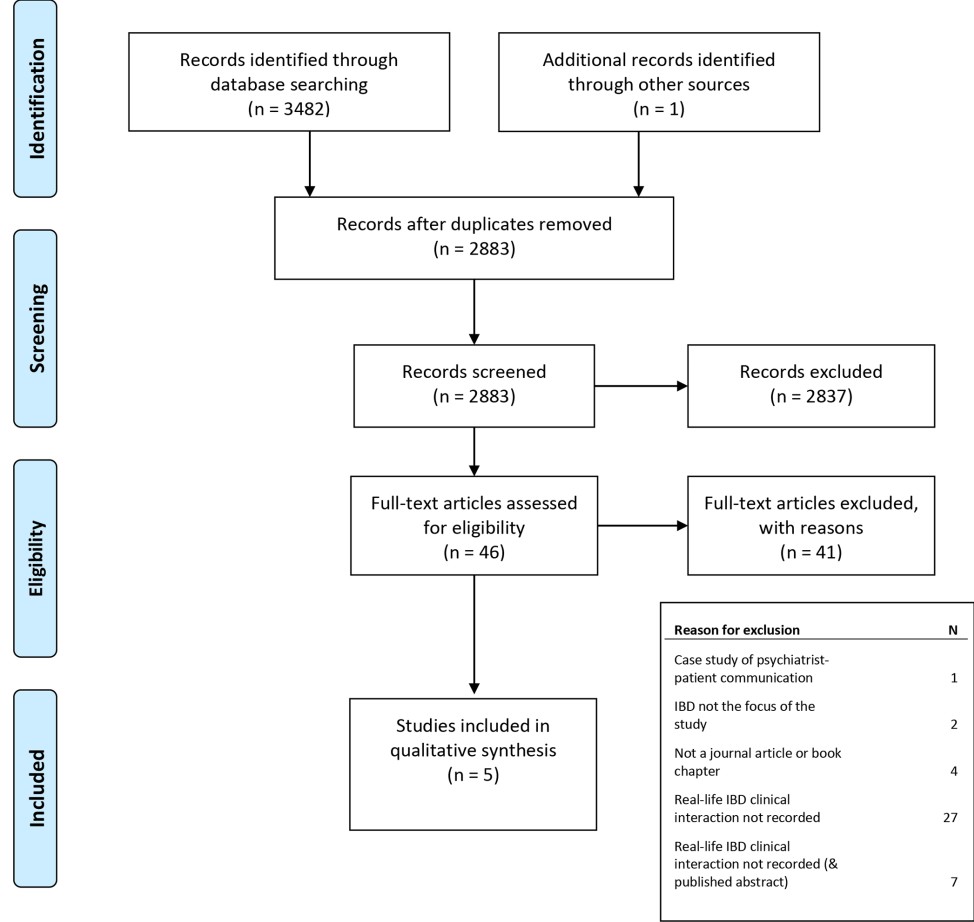

**Figure 1** The screening and inclusion process. IBD, inflammatory bowel disease.[49]

## Patient and public involvement

Patients or the public were not involved in the design and conduct of this systematic review.

## RESULTS

### Study selection and study characteristics

Of the 3482 search results identified as potentially relevant to our research question, 2883 unique publications were reviewed, and five papers from four individual studies fulfilled the inclusion criteria. Figure 1 illustrates the screening and inclusion process.

Included studies were conducted in six countries (The Netherlands, UK, Italy, Germany, France and USA) and in two institutional contexts (outpatient IBD nurse consultation, outpatient specialist consultation). Studies were published between 2008 and 2020. The characteristics of included studies are described in table 1 and the characteristics of the interaction data used in these studies are summarised in table 2.

### Risk of bias within studies

Following assessment of risk of bias (online supplemental tables 2A and 2B), all five publications were included in data synthesis.

Two publications[19 20] were mixed-methods correlational studies mainly examining selected predictive variables and

their relationship with medication adherence. The aim in these projects was to identify the predictors of medication intake behaviour which was facilitated through investigation of nurse–patient interaction. A related third publication was a mixed-methods study that investigated the communicative strategies employed by patients and IBD nursed to discuss online health information.[21] The fourth and fifth publications that were included in this review[22 23] were descriptive studies and detailed in terms of their linguistic analysis. Both studies combined analysis of recorded IBD specialist consultations with patient interviews. Rubin et al[22] conducted physician interviews, in addition to patient interviews, and investigated physician–patient alignment.

The analysis of the interactions in the two nurse-focused publications aiming to investigate predictors of medication intake behaviour was limited to determining the presence or absence of preselected pieces of medicines information. The analysis of clinical talk by Sanders and Linn[19] was limited to coding the interaction in terms of the presence of the words internet, Google(d), webpages, fora, online or any other internet-related words and the person initiating such discussion. Similarly, the analysis in the 2013 Linn et al study[20] included coding the data in terms of the presence or absence of certain pieces of information regarding prescribed medication. Other

**Table 1** Publication characteristics

| Publication | Research question(s) | Condition | Population and eligibility criteria | Country | Design | Measures/analytical framework | Whether part of a bigger study? |
|---|---|---|---|---|---|---|---|
| Sanders and Linn[19] 2018 | 1. How is talking about online information related to patient satisfaction, recall of medical information, and medication adherence?<br>2. How many patients seek online health information prior to their consultation?<br>3. How many patients and healthcare providers talk about online information and who initiates the discussion? | UC<br>CD | Patients (n=160)<br>Diagnosed with CD or UC<br>Starting azathioprine, methotrexate, adalimumab, infliximab, 6-mercaptopurine or 6-thioguanine<br>Able to speak and write Dutch<br>Nurses (n=8) | The Netherlands | Mixed methods | Online Medical Information Seeking Behaviour: Patients explicitly asked if they used the internet to search for medical information<br>Patient satisfaction: A 29-statement scale*<br>Recall of medical information: A structured telephone interview using an adapted version of the Netherlands Patient Information Recall Questionnaire (NPIRQ)†<br>Medication adherence: A single item self-report measure<br>Discussion of online medical information: Mention of (any combination of) the words *internet, Google(d), webpages, fora, online,* or any other internet-related words during the recorded consultation and the person initiating the discussion | Yes—part of a larger research project developing and testing a theoretical and evidence-based tailored multimedia intervention to improve medication adherence in patients with IBD |
| Linn et al[20] 2013 | What are the predictors of medication intake behaviour? (implied) | UC<br>CD | Patients (n=68)<br>▲ Diagnosed with CD or UC according to classical clinical, endoscopic, radiographic and/or pathological histological criteria as determined by an experienced gastroenterologist.<br>▲ Starting Azathioprine, 6-mercaptopurine, Infliximab, Methotrexaat, 6-thioguanine, or Adalimumab, and<br>▲ Fluent in Dutch<br>IBD nurses (n=8) | The Netherlands | Mixed methods | Content: An extensive observation checklist based on comparable studies in oncology<br>Immediate recall of medical information: An adapted version of The NPIRQ†<br>Delayed recall of medical information: An adapted version of the immediate recall questionnaire based on each individual consultation<br>Percentage of accurate recall: The ratio of the number of the accurately recalled items to the total number of items discussed coded by two coders using videotaped visits<br>Total recall score: The mean recall percentage per patient for immediate and delayed recall<br>Medication intake behaviour: A single item self-report measure | Yes—part of a larger study developing an intervention to tailor the communication to IBD patients' needs and barriers to medication intake behaviour |
| Linn et al[21] 2020 | How is online health information seeking discussed by patients and nurse practitioners? | UC<br>CD | Patients (Main study: n=165; current substudy: n=58)<br>Diagnosed with IBD<br>Starting immunosuppressive and biological therapy<br>IBD nurses (n=8) | The Netherlands | Mixed methods | Communicative strategies used to discuss online health information: Quantitative content analysis | Yes—part of a larger research project developing and testing a theoretical and evidence-based intervention to improve medication adherence in patients with IBD |

Continued

**Table 1** Continued

| Publication | Research question(s) | Condition | Population and eligibility criteria | Country | Design | Measures/analytical framework | Whether part of a bigger study? |
|---|---|---|---|---|---|---|---|
| Rubin et al[22] 2017 | ▲ Is there alignment between patients and physicians in terms of the impact of disease on the patient's QoL? (implied)<br>▲ What is perceived to be treatment success in patients' and doctors' view? (implied)<br>▲ In which visits the physician asked patients about QoL?<br>▲ Which patients were experiencing QoL and emotional impacts due to their disease and, of these, which patients discussed these impacts during the visit?<br>▲ Which patients demonstrated evidence of resigning themselves to active symptoms as acceptable (ie, 'a new normal' or 'learnt helplessness')?<br>▲ In which visits the physician interrupted the patient and the number of interruptions?<br>▲ In which visits the physician asked primarily closed-ended questions to assess symptoms?<br>▲ In which visits the physician conveyed a significant gap in their perception of conventional therapies for UC (eg, 5-ASA and corticosteroids) and advanced therapies (eg, immunomodulators and biologic therapies)?<br>▲ In which visits the physician advanced therapies were framed as a last resort and thus considered only if absolutely necessary?<br>▲ In which visits advanced therapies were discussed and where they were positioned in the treatment sequence?<br>▲ Which patients considered advanced therapies as more appropriate for more severe stages of disease than they believed they were experiencing? | UC | **US study**<br>Gastroenterologists (n=15)<br>▲ board-certified gastroenterologists, and spending at least 75% of their professional time in direct patient care<br>▲ be in full-time practice for 3–30 years<br>▲ primarily see patients in an office or private practice setting<br>▲ see at least 25 patients diagnosed with UC in a typical month<br>▲ have initiated biological therapy for at least 15% of their patients with UC<br>Patients (n=40)<br>▲ 18 years old or older<br>▲ Diagnosed with moderate-to-severe UC for ≥1 year<br>▲ At least one flare within the past year<br>▲ according to participating gastroenterologists<br>▲ Had previously received or were currently receiving therapy with 5-ASA and/or corticosteroids<br>▲ Fluent in English<br>▲ No cognitive impairment<br>**EU study**<br>Gastroenterologists (n=8)<br>1. Spending at least 60% of their time in direct patient care<br>2. Primarily treating adults<br>3. Main specialty being gastroenterology<br>4. Planning to discuss immunosuppressant or biological treatment with their patient at their next visit<br>5. In full-time practice for 3–30 years<br>6. Primarily seeing patients in an office or private practice setting<br>7. Seeing at least 25 patients diagnosed with UC in a typical month<br>8. Have initiated biologic therapy for at least 15% of their patients with UC<br>Patients (n=28)<br>1. 18 years old or older<br>2. Diagnosed with moderate-to-severe UC for ≥1 year<br>3. At least one flare within the past year<br>▲ according to participating gastroenterologists<br>4. Had previously received or were currently receiving therapy with 5-ASA and/or corticosteroids<br>5. No cognitive impairment | United States and Europe (Italy, Germany, France) | Mixed methods | Patient and physician attitudes toward and framing of immunomodulatory and biologic therapies:<br>EU: Qualitative analysis of visits and post-visit interviews<br>US:<br>▲ Number of of visits in which the physician conveyed a significant gap between conventional therapies and advanced therapies<br>▲ Number of visits in which the physician did not compare or discuss advance therapies<br>▲ Position of advanced therapies in the sequence of treatment in the visits in which they were discussed<br>▲ Number of visits in which gastroenterologists framed biological therapy as a last resort<br>▲ Number of visits in which it was clear that gastroenterologists had previously discussed advanced options<br>▲ Number of patients who were not currently or previously prescribed advanced therapies and who discussed advanced therapy options during the postvisit interview and their appraisal of advanced therapies during the interview<br>Patients and physicians' use and understanding of the term inflammation:<br>▲ Number of visits in which the term *inflammation* (or a variation thereof) was used by the physician<br>▲ Context in which the term *inflammation* was used by the physicians<br>▲ Number of visits in which the physician described what they meant by the term and explained the relevance of the term<br>▲ Number of visits in which the patient used the term *inflammation* (or variations)<br>▲ Number of interviews in which the patient used the term *inflammation* and context in which the patients used the term *inflammation*<br>Patient and physician description of treatment success:<br>Analysed in terms of absence of clinical symptoms (clinical remission) or absence of inflammation (endoscopic remission)<br>Further measures in the US study:<br>▲ Alignment between patients and physicians on QoL impacts: (1) complete alignment; (2) partial alignment; (3) no alignment<br>▲ Number of visits in which the physician asked patients about QoL or emotional impact of disease<br>▲ Number of patients who were experiencing QoL and emotional impacts due to their disease and number of patients who did not discuss or underplay these impacts during the visit<br>▲ Number of patients who demonstrated evidence of resigning themselves to active symptoms as acceptable (ie, 'a new normal' or 'learnt helplessness')<br>▲ Percentage of words spoken by the participants and the length of each visit<br>▲ Number of visits in which the physician interrupted the patient and the number of interruptions<br>▲ Number of visits in which the physician asked primarily closed-ended questions to assess symptoms | No |
| Radley et al[23] 2008 | How are the opportunities for the integration of patients' concerns enabled or hindered by the gastroenterologist? (implied) | UC<br>CD | **Main study**<br>Physicians (n=3), Patients (n=16)<br>**Reported case study**<br>▲ A woman patient (with her young daughter) with a registrar<br>▲ The same patient with a consultant<br>▲ A woman in her mid-30s, with a long history of UC who had become a mother for the first time 6 months previously (with her baby) with the consultant | UK | Qualitative | Patient expectations of the future consultation and patient reflection after the consultation: Episodic interviews<br>Interaction: the notion of *proto-story* | Yes—part of a project investigating how and when the concerns of IBD patients are included within the treatment protocols of their doctors |

Continued

## Table 1 Continued

| Publication | Research question(s) | Condition | Population and eligibility criteria | Country | Design | Measures/analytical framework | Whether part of a bigger study? |
|---|---|---|---|---|---|---|---|

*Linn AJ, van Weert JCM, van Dijk L, Horne R, Smit EG. The value of nurses' tailored communication when discussing medicines: Exploring the relationship between satisfaction, beliefs and adherence. Journal of Health Psychology. 2016;21(5):798-807.
†Jansen J, van Weert J, van der Meulen N, van Dulmen S, Heeren T, Bensing J. Recall in older cancer patients: measuring memory for medical information. Gerontologist. 2008;48(2):149-57.
CD, Crohn's disease; IBD, inflammatory bowel disease; QoL, quality of life; UC, ulcerative colitis.

than the demographic data, no contextual information was collected in the two publications. Nor were the analyses accompanied by examples of interactions. However, since the collected data and the processes for analysing qualitative data were relevant to address the research questions (refer to MMAT analysis), these two publications were included in the review. The analysis of talk in in the third IBD nurse communication study included in this review was more detailed than in the two earlier studies. This third study used grounded theory and quantitative content analysis to code the communicative data. Qualitative findings were quantified and accompanied by examples of single conversational turns or excerpts from single turns that instantiated them.

The studies that were focused on IBD specialist consultations provided examples from their textual data to support the findings, however, this was confined to single conversational turns or excerpts from single turns in Rubin et al.[22] Radley et al[23] included extra contextual data such as the consultant's level of experience and information on the personal life of the patient in their analysis, and included evidence from excerpts of gastroenterologist–patient interactions that illustrate patterns such as turn-taking and how topics are initiated and responded. However, these excerpts were taken from three consultations only.

None of the five studies provided sample size justification or mentioned use of a unit of analysis in their analysis. Only Linn et al[20] and Linn et al[21] determined interobserver reliability for text coding.

### Results: nurse–patient communication

The majority of the nurse–patient communication studies in this review[19 20] focused on the effect of communicating information regarding prescribed medicines (ideational metafunction) on patient medication intake behaviour. Only one recent study in our dataset looked at the dynamics of the nurse–patient relationship (interpersonal metafunction) and no study has looked at the structure of nurse-led consultations or the continuity of dialogue and how topics introduced by patients are maintained or ignored (textual metafunction).

Figure 2 summarises the findings of those publications in terms of the characteristics of nurse–patient interactions and their effects on patient outcomes and categorises these findings in terms of the linguistic function they address.

### Results: gastroenterologist–patient communication

Studies that investigated gastroenterologist–patient communication were different from those that investigated nurse–patient communication in their focus and objectives. The studies mainly focused on the interpersonal relationship between the gastroenterologist and the patient and the treatment decision-making process with the aim to reduce the distance between the patient and the consultant.

**Table 2** Characteristics of the interaction data

| Paper | Year | Country | Setting | No of sites | Corpus size | Average length of consultations | Whether series or one off | Participants | Patient demographics |
|---|---|---|---|---|---|---|---|---|---|
| Sanders and Linn 2018 [19] | 2018 | The Netherlands | Nurse–patient prescribing consultation—hospital | 6 | 165 videorecorded consultations (5 excluded from analysis due to lack of demographic data): 160 consultations | Not stated | One off | Nurses (n=8) Patients (n=160) | Gender (N): Female: 92, Male: 68 Age M (SD): 43.10 (15.33) Condition (N): CD: 101, UC: 49, other: 10 Diagnosed in years M (SD): 11.61 (10.55), Range: 1.5–47.1 Educational level: Low: 36, Moderate: 61, High: 63 |
| Linn et al 2013 [20] | 2013 | The Netherlands | Nurse–patient prescribing consultation—hospital | 6 | 59 videorecorded and nine audio-recorded consultations (68 consultations) | 29.5 min (SD=8.5) | One off | Nurses (n=8) Patients (n=68) | Gender (N): Female: 42, Male: 26 Age M (SD): 40.5 (14.9) Condition*: CD: 54, UC: 13, Unknown:1 Diagnosed in years M (SD): 9.6 (10.3) Education level: Low: 18, Moderate: 24, High: 26 Living arrangements: Alone: 16, With partner: 16, With partner and child(ren) 16, With Child(ren): 8, Other: 12 Employed: yes: 51 Ethnicity: Dutch: 62 |
| Linn et al [21] | 2020 | The Netherlands | Nurse–patient prescribing consultation—hospital | 6 | Main study: 165 recorded consultations Current sub-study: 58 recorded consultations | Not stated | One off | Nurses (n=8) Patients (n=58) | Gender (N): Female: 33, Male: 25 Age M: 48 Condition (N): CD: 34, UC: 20, Unknown: 2 Diagnosed in years M: 10 Educational level: Low: 6, Moderate: 22, High: 28, Unknown: 2 |
| Rubin et al 2017 [22] | 2017 | US and Europe (Italy, Germany, France) | outpatient clinics–community-based practices | Not stated | US study: 40 visits EU study: 28 visits | 15 min (US research) | One off | US study: Gastroenterologists (n=15) Patients (n=40) EU study: Gastroenterologists (n=8, Italy: 5, Germany: 2, France: 1) Patients (n=28) | US study: Gender: Female: 21, Male: 19 Age M (range): 49 (20–83) EU study: Gender: Female: 11, Male: 17 Age: 18–34 years old: 4 patients 35–54 years old: 14 patients 55–74 years old: 4 patients Unavailable: 6 patients |
| Radley et al 2008 [23] | 2008 | UK | Outpatient clinic - hospital | 1 | 3×16 videorecorded consultations (assumed information, not explicitly stated in the article) | Not stated | Series (three consultations from each patient) | Physicians (n=3) Patients (n=16) | Not stated |

CD, Crohn's disease; EU, Europe; UC, ulcerative colitis; ;US, United States.

■ Interpersonal   ■ Experiential   ■ Textual

**Nurse-patient clinical communication**

**(20)**

**Direct reference to online health information:** In 46.8% of the 95 consultations (n=44), online health information was directly referenced. Overall, reference to online health information was almost evenly initiated by patients (n=33; 55%) and health-care providers (n=27; 45%).

Patients who directly refered to online health information during consultation were more satisfied with the consultation (mean=0.86; SD=.61) compared to patients who searched online but did not talk about it (mean=1.30; SD=1.13, Mdiff=-0.43, p=.041, d=0.48).

There was no significant relation between reference to online health information and recall of medical information (Mdiff=0.12, p=0.310).

There was no significant direct relation between patient satisfaction and the recall of medical information (Mdiff=-0.13, p=0.270)
For patients referring to online health information, higher satisfaction with the consultation related to higher recall of medical information, compared to not discussing it (Mdiff=0.32, p=0.054).

**(21)**

**Medical information discussed (% of consultations in which each category was discussed):** Purpose of the medication (89.7%); Name of medication (98.5%); Duration of treatment (48.5%); Frequency of administration (97.1%); Influence of the medication on the immune system (88.2%); When to expect an effect from the medication (80.9%); The need for blood monitoring (80.9%); Side effects (100%); When to contact the nurse (94.1%); The possibility of experiencing side effects (47.1%); The impact of the medication on the patients' daily life (79.4%); medication intake behaviour advice (44.1%)

**Medical information recalled (%):** Purpose of the medication (32.7%); Name of medication (83.6%); Duration of treatment (59.8%); Frequency administration (85.9%); Influence of the medication on the immune system (69.2%); When to expect an effect from the medication (66.8%); The need for blood monitoring (63.6%); Side effects (26.7%); When to contact the nurse (39.2%); The possibility of experiencing side effects (78.8%); The impact of the medication on the patients' daily life (5.5%); Medication intake behaviour advice (15%).

Delayed recall of medical information was significantly related to self-reported medication intake behavior (β=0.37, p=0.007). Patients with lower recall scores three weeks after the consultation rated themselves as less adherent than the patients with higher recall scores.

**(22)**

**Discussions about online health information-seeking initiated**

**Patient's initiation strategies**
Making a general statement (n=19, 43 %)
Raising concerns (n=8, 18 %)
Asking questions (n=7, 16 %)
Citing specific information (n=5, 11 %)
Dismissing (n=5, 11 %)
**Nurses reaction strategies**
Taking patients' online health information-seeking seriously (n=20, 43 %)
Affirming patients' beliefs (n=11, 24 %)
Ignoring (n=6, 13 %)
Minimal encouragement (n=5, 11 %)
Correcting (n=5, 11 %)

**Discussions about online health information-seeking initiated by nurses:**
**Nurse's initiation strategies**
Exploring (n=21, 50 %)
Warning (n= 13, 31 %)
Encouraging (n=8, 19 %)
**Patient's reaction strategies**
Disclosing (n=15, 40 %)
Reacting minimally (n=9, 24 %)
Expressing concerns (n=8, 21 %)
Substantive reaction (n=6, 16 %)

| Nurse | Recorded consultations, n | Consultations in which the internet is discussed, n (%) | Nurse initiations, n (%) |
|---|---|---|---|
| 1 | 17 | 4 (23%) | 1 (25%) |
| 2 | 10 | 1 (10%) | 1 (100%) |
| 3 | 16 | 3 (18%) | 0 (0%) |
| 4 | 15 | 6 (40%) | 5 (83%) |
| 5 | 15 | 8 (53%) | 7 (88%) |
| 6 | 50 | 20 (40%) | 16 (80%) |
| 7 | 32 | 14 (44%) | 5 (36%) |
| 8 | 25 | 1 (4%) | 0 (0%) |

**Figure 2** Findings of articles investigating nurse–patient communication categorised by linguistic metafunction.

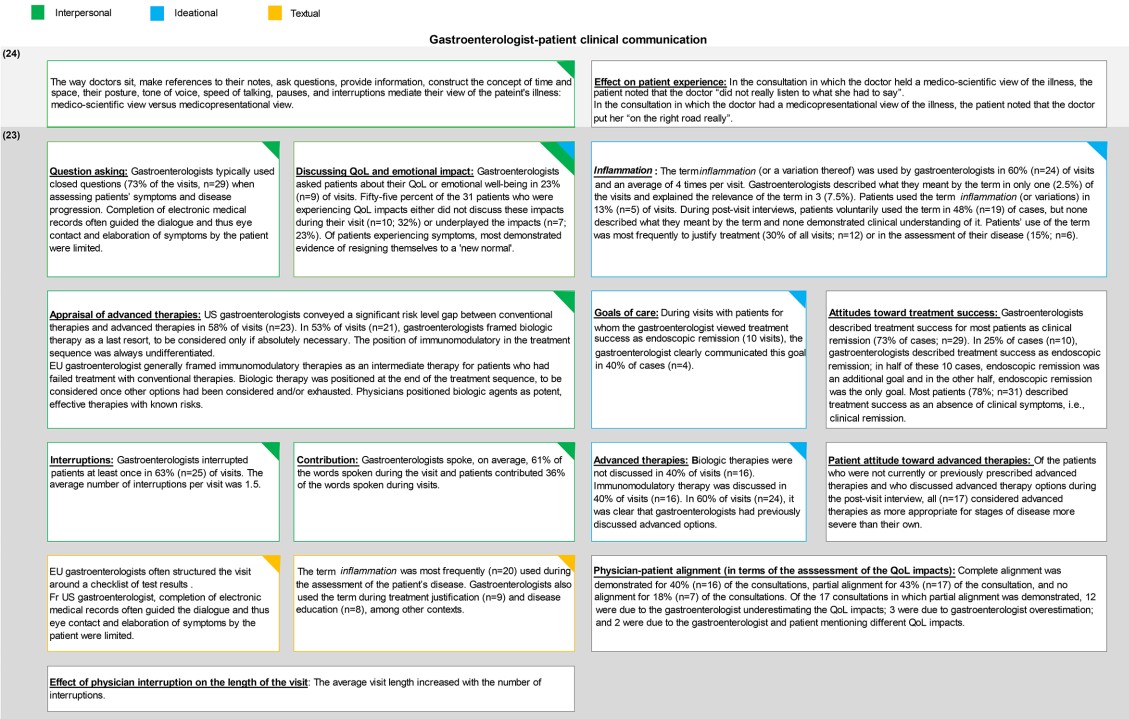

**Figure 3** Findings of articles investigating gastroenterologist-patient communication categorised by linguistic metafunction. EU, Europe; QoL, quality of life; US, United States

Figure 3 summarises the findings of the two articles that described the characteristics of the gastroenterologist–patient communication and categorises these findings in terms of the linguistic function they investigate.

## DISCUSSION
### Nurses as transmitters of information
Historically, the IBD health communication literature identified nurses only as transmitters of information about new medications, assuming their interactions with patients to have only an ideational function. Using this methodology, data relating to turn-taking, boundaries (who initiates a topic and who closes a topic), evaluation and attitudes, elaboration or engagement are not captured. The interpersonal function of the nurse–patient communication in IBD has more recently started to attract the attention of IBD researchers. The 2020 Linn et al study looked at the way online health information-seeking is discussed by patients and nurse practitioners during prescription consultations and investigated their interactions from an interpersonal perspective.[21]

Nurses are at the forefront of IBD healthcare, providing medical and psychosocial care (including but not limited to provision of information) via IBD advice-lines and virtual clinics and are at times the mediator or gatekeeper between the patient and the gastroenterologist.[24–29] Understanding the dynamics of the IBD nurse–patient interaction as a bidirectional process that both responds to the context at hand and reshapes it will help develop effective communication strategies and identify communicative risks. In achieving this understanding, a robust account of variation in nurse–patient interaction is crucial since small changes in ways of speaking can produce big differences in how a context is experienced and interpreted by patients. The current literature, seemed to use 'common sense' rather than evidence, to provide nurse–patient communication recommendations. For example, Linn et al's[20] suggestion that signposting and categorising information can improve recall was not grounded in the findings of their study. These recommendations seem to be hypotheses that would need testing.

### Discussion of online health information
Sanders and Linn showed that only about half the patients who had searched for medical-related online information prior to the nurse prescription consultation discussed or had the opportunity to discuss that information.[19] The 2020 Linn et al paper revealed that patients and nurses were equally likely to initiate the discussion of online information-seeking; and that they did so often by using a general strategy such as making a general statement on the part of the patient and exploring whether (and what) the patient searched online before the consultation on the part of the nurse.[21] Nurses, however, varied in terms of broaching the discussion of online health information with some initiating more frequently than others, according to this study. Barriers to discussing online health information are known from studies outside IBD to include clinician's resistance towards discussing such information, patient concerns surrounding disapproval by the clinician or fear of embarrassment.[30] It has also been

shown outside IBD that men are more likely than women to have a conversation regarding online information with physicians.[31] Internet health information seeking and discussing such information during the consultation can empower patients and give them a sense of control and confidence when talking to their clinician.[30] It can also help them gain greater clarity, orientation, and certainty regarding their own health and lead to higher patient satisfaction as shown in this review, so it seems important to increase opportunities for such discussion. Research has shown that the clinician's communication skills and their open response to a less hierarchised relationship as a result of a patient discussing online health information is important in forming a positive relationship with the patient.[30]

## Gastroenterologist–patient communication

Gastroenterologist–patient communication has been conceptualised around the interpersonal relationship between the gastroenterologist and the patient and its role in the decision-making process. However, the focus of the literature is on the gastroenterologist's contribution to the dialogue with less emphasis on contributions from the patient. Furthermore, little information is provided on the structure of the IBD specialist consultations. What is known, in terms of the structure is that gastroenterologists tended to use medical records to guide the structure of the consultation.[22 23] However, alternative options have not been elaborated in the literature specific to gastroenterology and we do not know the extent to which specialists in this field come into contact with alternative models used in other contexts of care.

## The voice of the lifeworld versus the voice of medicine

Mishler's concept of *voice* has been used in many health communication research contexts including IBD (by Radley *et al*). Mishler used the concept to 'specify relationships between talk and speakers' underlying frameworks of meaning[32]' and distinguished between the *voice of medicine* and the *voice of the lifeworld*. The *voice of medicine* represents 'the technical-scientific assumptions of medicine' and the *voice of the lifeworld* represents 'the natural attitude of everyday life', according to Mishler[32] (p. 13). Radley *et al* argued that these voices are not mutually exclusive.[23] However, the researchers showed that by introducing the *voice of the lifeworld* or what they refer to as the *medico-presentational* regime, the gastroenterologist managed to create a link between IBD as a disease and the patient experience of having IBD and facilitate what can be referred to as shared decision making.

Nevertheless, the existing IBD communication literature seems to suggest that the dominant voice in the IBD specialist consultations and potentially IBD nurse consultations is the *voice of the medicine*. What strengthens this hypothesis for IBD nurse consultations is the fact that only about half the patients who had searched for medical-related online information prior to the nurse prescription consultation discussed or had the opportunity to

discuss that information.[19] In IBD specialist consultations, this is construed through the gastroenterologists' closed-ended questions when assessing patients' symptoms and disease progression and their interruptions and absence of explicit negotiations of treatment goals during consultations. The dominance of the *voice of the medicine* in IBD specialist consultations has also been demonstrated by the structure of the visits centring around electronic medical records and a checklist of test results.[22] This is despite electronic medical records having the potential to be designed[33] or used[34 35] in a way that facilitates a patient-centred clinical encounter. What further strengthens the claim that the *voice of the medicine* is generally the dominant voice in IBD specialist consultations is that gastroenterologists in the USA asked patients about the impact of IBD on their quality of life or emotional well-being in less than a quarter of visits.[22] More than half the patients whose quality of life had been impacted by their IBD either did not discuss these impacts with their physician or underplayed the impacts. Furthermore, complete physician–patient alignment in terms of assessment and understanding of the quality of life impacts was present in less than half the cases. In agreement with these findings are the results of the European Federation of Crohn's and Ulcerative Colitis Associations patient survey in 2007 which showed that less than half of the physicians asked their patients with IBD about their quality of life and less than half of the patients initiated a conversation with their physicians about quality of life.[36] Furthermore, research suggested that patients discussed quality of life only if asked directly[2] and were not as comfortable with discussing emotional concerns as their gastroenterologists believed them to be,[37] which highlights the value of establishing the *voice of the lifeworld* in conversations with patients with IBD.

Importantly, orienting to the lifeworld in clinical consultation does not mean merely allowing patients to prioritise topics but also means that patients' evaluations of the relative risks and benefits of different management strategies and the way these fit with their values must also be negotiated. The opportunity to negotiate treatment options in the context of patient values allows patients to potentially reason through their resistance towards the take-up of recommended treatment, which is generally treated as simply a problem (eg, by Sanders and Linn, as shown in this review) but which other researchers describe as 'rational action' on the part of patients.[38] An enhanced critical awareness of this literature would benefit IBD research and practice.

From a research design perspective, orienting to the lifeworld could arguably mean establishing research that places the patient at the centre of a network of clinical relationships and information sources to do with managing their disease. Sanders and Linn have gone some way towards this by exploring how information outside the clinic is considered in the context of clinical consultation. This approach could be extended to augment our understanding of the roles of nurses, specialists, general

practitioners, friends, family, internet information pages, internet support groups and how people with IBD negotiate such networks of information and values.

## Goals of treatment, patient knowledge and the representation of advanced therapies

The existing IBD communication literature demonstrated that treatment goals were not explicitly negotiated by the gastroenterologist with their patient in the consultation. In addition, patients did not seem to be equipped with the necessary knowledge to be able to effectively negotiate the goals of treatment and make a decision. In particular, Rubin *et al* observed that even though the term *inflammation* was frequently used by the patients and the gastroenterologists in the consultations, patients did not demonstrate clinical understanding of the term and gastroenterologists did not explain its meaning. This is supported by the current general literature of IBD which suggests that patients with IBD do not have as much disease-related knowledge as they need to manage their condition in an ideal way.[37 39–42]

When transparency regarding goals of treatment as well as patients' disease-related knowledge are suboptimal, shared decision making is compromised. In this context, the way gastroenterologists represent different therapy options in their interactions with patients (intentionally or unintentionally) may be the only available resource for patient decision making about IBD treatment. However, research has shown that the way clinicians subconsciously represent options can be different from what they think is best.[43] This increases the risk of patients choosing an option that disagrees either with their own preferences or the clinical recommendations. A possible avenue for further research is to investigate and compare gastroenterologists' representation of treatment options in the consultation with their opinion about the best treatment option, assessed in a postvisit interview.

## The importance IBD-specific health communication research

Earlier in this paper, we argued that it is important to study the details of how clinical communication takes place in the context of IBD and that drawing only on generic principles of healthcare communication or research involving patients with other health conditions is not adequate for optimising IBD care delivery. The importance of patient engagement, shared decision making and patient-centred care in IBD has been emphasised by researchers, clinicians and institutions for a long time. Different IBD standards recommend the implementation of general principles from health communication research including supporting patients to exercise choice between treatments and follow-up care models, information provision, and education (eg, Australian and European Standards[44 45]). Despite that, this review of the very limited literature on observed actual IBD communication suggests that IBD specialist care is not as patient-centred as it could be and shared decision making does not always occur in the consultations. The review also highlighted

an important gap in knowledge, confirming that there is currently a limited capacity for evidence-based communication in IBD because there is only rather limited evidence.

Researchers and practitioners can make hypotheses about how results on communication in other diseases transfer to IBD, however, those hypotheses need to be examined and supported by evidence. In the absence of evidence resulting from the systematic study of clinical communication in IBD, it cannot be certain whether certain recommendations would have the same effect in the specific context of IBD. It is also possible to miss issues that might be discovered in IBD for the first time or those that might be concentrated in IBD but affect other disease groups too. As such, researchers need to team up with the community of IBD practitioners and patients to identify and address the communication issues experienced by them instead of simply doing research on them.

## Summary of findings

This systematic review reveals that there has been insufficient attention to naturally occurring consultation dialogue in the research on IBD. The review findings suggest that historically IBD nurses have been identified in the IBD literature solely as transmitters of information about new medications. However, this has started to change recently. There is a new strand of research in the IBD literature that acknowledges and investigates the interpersonal meanings exchanged in IBD nurse consultations. Findings so far suggest the importance of nurses eliciting discussion about what kind of online information patients use to help them understand IBD medications and to situate illness and treatment within their lives. Turning to physician–patient communication, our review suggests that IBD care remains unhelpfully disease-centred because of a tightly structured consultation around the consultant's agenda including the completion of electronic medical records and the review of test results and because the patient's quality of life is not fully addressed during the consultation. In addition, shared decision making in IBD specialist consultations seems to be compromised due to the lack of explicit negotiation around goals of therapy. Information on the different structural components of IBD nurse or specialist consultations is not available in the literature.

## Limitations

The findings of this systematic review, however, should be interpreted in the light of its limitations. First, the findings could not be integrated to produce cumulative evidence due to the diverse range of included studies in terms of context, theoretical underpinnings and data types. However, analysis of findings in terms of the linguistic metafunction(s) they address, resulted in the generation of new evidence which revealed the gaps in IBD communication research and provided suggestions for further research as discussed throughout this review. In addition, where possible in the discussion section, links were made

between the existing findings to generate new knowledge. A second limitation was that we excluded reports published by abstract only. There were three relevant published abstracts that were excluded from this review. One was related to an included publication.[22 46] One was a 2010 descriptive study of interactions between gastroenterologists and patients with Crohn's disease regarding treatment with biologic therapy.[47] Information on any potential eligible publication related to this project was not available at the time of submitting this review. The other excluded published abstract reported the results of an ongoing observational study of 102 IBD clinic consultations conducted by 24 IBD clinicians (10 consultants, 10 IBD nurses and 4 trainees) with the aim to explore variability in the assessment and recording of clinical IBD outcomes during routine practice in England.[48] Further work from this project is currently under review at the time of writing this systematic review, according to the first author. The other limitation of this review was that we considered only articles that were in English. However, this limitation did not seem to influence the results of the review. While non-English publications were excluded at the level of database search for PsycINFO and EMBASE, they were included in the database search and later excluded manually by the reviewers for the other five databases. This included 103 papers in 16 languages other than English. None used real-world clinician–patient interaction data in their analysis according to our review of the English translated abstracts. These publications are not reflected in the number of publications reported in this project.

## CONCLUSIONS

Further research needs to be conducted to understand the dynamics and details of communication between IBD clinicians (including nurses, consultants and other members of the multidisciplinary team) and patients (including adults, adolescents, preconception and pregnant women, etc) and its effect on patient outcomes. Crucial to such research will be the systematic analysis of clinical interaction with a focus on contributions from all parties including the patient, the consultant, IBD nurses, the dietitian and even family or friends attending with the patient, rather than profiling only what clinicians do. This kind of approach allows interactions between the patient and the gastroenterologist, for example, to be interpreted in the context of the patient's interactions with other members of the multidisciplinary team. Such research acknowledges that the patient is the centre of a dense network of meanings and relations around their disease and draws on a functionally oriented model of language to explore these meanings and relations.

**Author affiliations**
[1]Gastroenterology and Liver Research Group, Ingham Institute for Applied Medical Research, Liverpool, New South Wales, Australia
[2]South Western Sydney Clinical School, UNSW Medicine, University of New South Wales, Sydney, New South Wales, Australia
[3]Gastroenterology Department, Liverpool Hospital, Liverpool, New South Wales, Australia
[4]Department of Linguistics, Faculty of Medicine, Health and Human Sciences, Macquarie University, Sydney, New South Wales, Australia
[5]English Language & Linguistics, Faculty of Law Humanities and the Arts, University of Wollongong, Wollongong, New South Wales, Australia

**Contributors** NK, ARM and AL conceived the idea of this systematic review project. NK led the project, developed the protocol, conducted the review and prepared the first draft of this manuscript with constructive feedback from ARM, AL, SC, RK and A-JW on the design of the protocol, the review process and the manuscript. RK and AL conducted the title and abstract screening as second reviewers. ARM provided expert advice and guidance throughout the project and on the manuscript as the senior health communication researcher on this project.

**Funding** The authors have not declared a specific grant for this research from any funding agency in the public, commercial or not-for-profit sectors.

**Competing interests** NK has received grant and educational support from Janssen and Ferring. ARM and AL have received grant support from Janssen. RK has received research and educational support from Pfizer, Abbvie, Takeda and Janssen. AJW has received Honoraria and/or research support from Takeda, Ferring, Janssen, Pfizer and Abbvie. SJC has received honoraria for Advisory Board participation, speaker fees, educational support and/or research support from: Abbvie, Aspen, BMS, Celgene, Celltrion, Chiesi, DrFalk, Ferring, Fresenius Kabi, Gilead, Janssen, MSD, Novartis, Pfizer, Takeda, Vifor, Agency for Clinical Innovation, Gastroenterological Society of Australia, Medical Research Future Fund, The Leona M and Harry B Helmsley Charitable Trust, South Western Sydney Local Health District.

**Patient consent for publication** Not required.

**Ethics approval** Ethics reviews and approval were not required for this systematic review as no primary data were collected.

**Provenance and peer review** Not commissioned; externally peer reviewed.

**Data availability statement** Data are available on reasonable request. The data underlying this article will be shared on reasonable request to the corresponding author.

**ORCID iDs**
Neda Karimi http://orcid.org/0000-0002-2841-637X
Ria Kanazaki http://orcid.org/0000-0002-8866-1078
Annabelle Lukin http://orcid.org/0000-0003-1023-7207
Alison Rotha Moore http://orcid.org/0000-0002-9697-5747
Susan Connor http://orcid.org/0000-0001-5606-0270

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
