## [Reviewer comments · BMJ Open]

ARTICLE DETAILS

TITLE (PROVISIONAL)	Clinical communication in inflammatory bowel disease: a systematic review of the study of clinician-patient dialogue to inform research and practice
AUTHORS	Karimi, Neda; Kanazaki, Ria; Lukin, Annabelle; Moore, Alison; Williams, Astrid-Jane; Connor, Susan

VERSION 1 – REVIEW

REVIEWER	Johnston, Amy University of Ottawa, School of Epidemiology and Public Health
REVIEW RETURNED	24-Mar-2021

GENERAL COMMENTS	Thank you for the opportunity to review this interesting article. I only have one minor suggestion: Page 7; 2 paragraphs beginning on line 33: These 2 paragraphs provide a general overview of the study design along with findings specific to ROB. To improve the flow, I would suggest providing a general summary of the studies under one heading followed by a more focused discussion of ROB below.
---

REVIEWER	Limdi, Jimmy The Pennine Acute Hospitals NHS Trust, Gastroenterology
REVIEW RETURNED	21-May-2021

GENERAL COMMENTS	This paper addresses an important unmet need with respect to the quality of consultations between patients with IBD and their clinicians and their effect on clinical outcomes. Given current limitations with research in this sphere, the findings may (or should) provide impetus for further work. You should highlight the key findings of your study at the end before stating your limitations. Also areas of further research need to be clearly defined. The wording appears somewhat ambiguous here to a clinician.
---

VERSION 1 – AUTHOR RESPONSE

Reviewer 1	Page 7; 2 paragraphs beginning on line 33: These 2 paragraphs provide a general overview of the study design along with findings specific to ROB. To improve the flow, I would suggest providing a general summary of the studies under one heading followed by a more focused discussion of ROB below.	Revised (see pages 5-6 in the tracked revised manuscript)
------------	---	---

Reviewer 2	You should highlight the key findings of your study at the end before stating your limitations.	The key findings now summarised at the end of the 'Discussion' section (see page 11 in the tracked revised manuscript).
	Also areas of further research sed to be clearly defined. The wording appears somewhat ambiguous here to a clinician.	Areas of further research elaborated (see page 12 in the tracked revised manuscript).